# Case-based Reasoning for Better Generalization in Textual Reinforcement Learning

**Mattia Atzeni**
IBM Research, EPFL
atz@zurich.ibm.com

**Shehzaad Dhuliawala**
ETH Zürich
shehzaad.dhuliawala@inf.ethz.ch

**Keerthiram Murugesan**
IBM Research
keerthiram.murugesan@ibm.com

**Mrinmaya Sachan**
ETH Zürich
mrinmaya.sachan@inf.ethz.ch

## Abstract

Text-based games (TBG) have emerged as promising environments for driving research in grounded language understanding and studying problems like generalization and sample efficiency. Several deep reinforcement learning (RL) methods with varying architectures and learning schemes have been proposed for TBGs. However, these methods fail to generalize efficiently, especially under distributional shifts. In a departure from deep RL approaches, in this paper, we propose a general method inspired by case-based reasoning to train agents and generalize out of the training distribution. The case-based reasoner collects instances of positive experiences from the agent's interaction with the world in the past and later reuses the collected experiences to act efficiently. The method can be applied in conjunction with any existing on-policy neural agent in the literature for TBGs. Our experiments show that the proposed approach consistently improves existing methods, obtains good out-of-distribution generalization, and achieves new state-of-the-art results on widely used environments.

## 1 Introduction

Text-based games (TBGs) have emerged as key benchmarks for studying how reinforcement learning (RL) agents can tackle the challenges of grounded language understanding, partial observability, large action spaces, and out-of-distribution generalization (Hausknecht et al., 2020; Ammanabrolu & Riedl, 2019). While we have indeed made some progress on these fronts in recent years (Ammanabrolu & Hausknecht, 2020; Adhikari et al., 2020; Murugesan et al., 2021b;a), these agents are still very inefficient and suffer from insufficient generalization to novel environments. As an example, state-of-the-art agents require 5 to 10 times more steps than a human to accomplish even simple household tasks (Murugesan et al., 2021b). As the agents are purely neural architectures requiring significant training experience and computation, they fail to efficiently adapt to new environments and use their past experiences to reason in novel situations. This is in stark contrast to human learning which is much more robust, efficient and generalizable (Lake et al., 2017).

Motivated by this fundamental difference in learning, we propose new agents that rely on case-based reasoning (CBR) (Aamodt & Plaza, 1994) to efficiently act in the world. CBR draws its foundations in cognitive science (Schank, 1983; Kolodner, 1983) and mimics the process of solving new tasks based on solutions to previously encountered similar tasks. Concretely, we design a general CBR framework that enables an agent to collect instances of past situations which led to a positive reward (known as cases). During decision making, the agent retrieves the case most similar to the current situation and then applies it after appropriately mapping it to the current context.

The CBR agent stores past experiences, along with the actions it performed, in a case memory. In order to efficiently use these stored experiences, the agent should be able to represent relevant contextual information from the state of the game in a compact way, while retaining the property

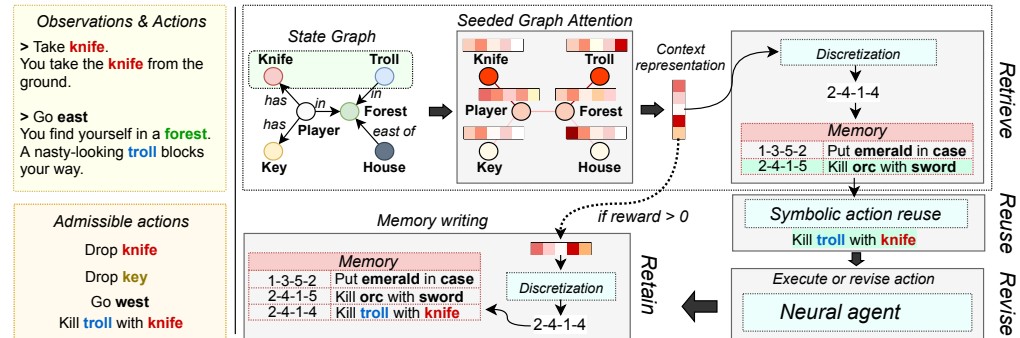

**Figure 1:** Overview of the approach and architecture of the CBR agent. A memory stores actions that have been used in previous interactions. The context of the game is learned from the state knowledge graph using a graph attention mechanism. Actions are retrieved from the memory based on this context representation and mapped to the current state. If no valid action is obtained using CBR, the algorithm falls back to a neural agent.

that contexts that require similar actions receive similar representations. We represent the state of the game as a knowledge graph (Ammanabrolu & Hausknecht, 2020) and we address these challenges by utilizing *(a)* a seeded message propagation that focuses only on a subset of relevant nodes and *(b)* vector quantization (Ballard, 2000) to efficiently map similar contexts to similar discrete representations. Vector quantization allows the model to significantly compress the context representations while retaining their semantics; thereby, allowing for a scalable implementation of CBR in RL settings. An illustration of the framework is shown in Figure 1.

Our experiments show that CBR can be used to consistently boost the performance of various on-policy RL agents proposed in the literature for TBGs. We obtain a new state-of-the-art on the *TextWorld Commonsense* (Murugesan et al., 2021b) dataset and we achieve better or comparable scores on 24 of the 33 games in the *Jericho* suite (Hausknecht et al., 2020) compared to previous work. We also show that CBR agents are resilient to domain shifts and suffer only marginal drops in performance **(6%)** on out-of-distribution settings when compared to their counterparts **(35%)**.

## 2 PRELIMINARIES

**Text-based games.** Text-based games (TBGs) provide a challenging environment where an agent can observe the current state of the game and act in the world using only the modality of text. The state of the game is hidden, so TBGs can be modeled as a Partially Observable Markov Decision Process (POMDP) $(\mathcal{S}, \mathcal{A}, \mathcal{O}, \mathcal{T}, \mathcal{E}, r)$, where $\mathcal{S}$ is the set of states of the environment of the game, $\mathcal{A}$ is the natural language action space, $\mathcal{O}$ is the set of observations or sequences of words describing the current state, $\mathcal{T}$ are the conditional transition probabilities from one state to another, $\mathcal{E}$ are the conditional observation probabilities, $r : \mathcal{S} \times \mathcal{A} \to \mathbb{R}$ is the reward function, which maps a state and action to a scalar reward that the agent receives.

**Case-based reasoning.** Case-based reasoning (CBR) is the process of solving new problems based on the solution of previously seen similar problems. Generally, CBR assumes access to a memory that stores past problems (known as cases) and their solutions. When a new problem is encountered, CBR will *(i)* **retrieve** a similar problem and its solution from memory; *(ii)* **reuse** the solution by mapping it to the current problem; *(iii)* **revise** the solution by testing it and checking whether it is a viable way to address the new problem; and *(iv)* **retain** the solution in memory if the adaptation to the new problem was successful.

## 3 CASE-BASED REASONING IN REINFORCEMENT LEARNING

This section introduces our framework inspired by CBR for improving generalization in TBGs. Even though we situate our work in TBGs, it serves as a good starting point for applying CBR in more general RL settings. We consider an *on-policy* RL agent that, at any given time step $t$, has access to a memory $\mathcal{M}_t$, that can be used to retrieve previous experiences. The memory contains

key-value pairs, where the keys are a context representation of a game state and values are actions that were taken by the agent w.r.t to this context. As mentioned in Section 2, case-based reasoning can be formalized as a four-step process. We describe our proposed methodology for each step below. Algorithm 1 provides a detailed formalization of our approach.

**Retrieve.** Given the state of the game $s_t$ and the valid actions $\mathcal{A}_t$, we want to retrieve from the memory $\mathcal{M}_t$ previous experiences that might be useful in decision-making at the current state. To this end, for each admissible action $a_t \in \mathcal{A}_t$, we define a context selector $c_t = context(s_t, a_t)$. The context selector is an action-specific representation of the state, namely the portion of the state that is relevant to the execution of an action. We will explain later how the context selector is defined in our implementation. For each context $c_t$, we retrieve from the memory the context-action pair $(c_t^{\mathcal{M}}, a_t^{\mathcal{M}})$, such that $c_t^{\mathcal{M}}$ has maximum similarity with $c_t$. We denote as $\delta = sim(c_t, c_t^{\mathcal{M}}) \in [0, 1]$ the relevance score given to the retrieved action. Only actions $a_t^{\mathcal{M}}$ with a relevance score above a *retriever threshold* $\tau$ are retrieved from $\mathcal{M}_t$. We denote as $\mathcal{A}_t^{\mathcal{M}}$ the final set of action-relevance pairs returned by the retriever, as shown in Algorithm 1.

**Reuse.** The goal of the reuse step is to adapt the actions retrieved from the memory based on the current state. This is accomplished by a *reuse* function, that is applied to each retrieved action to construct a set $\tilde{\mathcal{A}}_t$ of candidate actions that should be applicable to the current state, each paired with a confidence level.

**Revise.** If any of the action candidates $\tilde{\mathcal{A}}_t$ is a valid action, then the one with the highest relevance $\delta$ is executed, otherwise a neural agent $\pi$ is used to select the best action $a_t^\star$. We denote with $r_t = r(s_t, a_t^\star)$ the obtained reward. Note that $\pi$ can be an existing agent for TBGs (Murugesan et al., 2021c;b; Ammanabrolu & Hausknecht, 2020).

---

**Algorithm 1:** CBR in Text-based RL

● **Retrieve**
Let $\mathcal{C}_t = \{context(s_t, a_t) \mid a_t \in \mathcal{A}_t\}$ be a set
 of context selectors for state $s_t$ at time step $t$
$\mathcal{A}_t^{\mathcal{M}} \leftarrow \emptyset$
**for** $c_t \in \mathcal{C}_t$ **do**
  Let $(c_t^{\mathcal{M}}, a_t^{\mathcal{M}}) =$
   $\arg\max_{(c_t^{\mathcal{M}}, a_t^{\mathcal{M}}) \in \mathcal{M}_t} sim(c_t, c_t^{\mathcal{M}})$
  Let $\delta = sim(c_t, c_t^{\mathcal{M}})$
  **if** $\delta > \tau$ **then**
   $\mid$ $\mathcal{A}_t^{\mathcal{M}} \leftarrow \mathcal{A}_t^{\mathcal{M}} \cup \{(a_t^{\mathcal{M}}, \delta)\}$
  **end**
**end**

● **Reuse**
Build a set of action candidates:
  $\tilde{\mathcal{A}}_t = \{reuse(a_t^{\mathcal{M}}, s_t, \delta) \mid$
    $(a_t^{\mathcal{M}}, \delta) \in \mathcal{A}_t^{\mathcal{M}}\}$

● **Revise**
**if** $\mathcal{A}_t \cap \tilde{\mathcal{A}}_t \neq \emptyset$ **then**
  $\mid$ Let $a_t^\star, \delta^\star = \arg\max_{\tilde{a}_t, \delta \in \tilde{\mathcal{A}}_t} \delta$
**else**
  $\mid$ $a_t^\star = \arg\max_{a_t \in \mathcal{A}_t} \pi(a_t | s_t)$
**end**
Let $r_t = r(s_t, a_t^\star)$ be the reward obtained at
 time step $t$ by executing action $a_t^\star$

● **Retain**
Let $c_t^\star = context(s_t, a_t^\star)$ be the context of
 action $a_t^\star$
$\mathcal{T} = \{(c_t^\star, a_t^\star), \ldots, (c_{t-m+1}^\star, a_{t-m+1}^\star)\}$
**if** $r_t > 0$ **then**
  $\mid$ $\mathcal{M}_{t+1} \leftarrow \mathcal{M}_t \cup retain(\mathcal{T})$
**end**

---

**Retain.** Finally, the retain step stores successful experiences as new cases in the memory, so that they can be retrieved in the future. In principle, this can be accomplished by storing actions for which the agent obtained positive rewards. However, we found that storing previous actions as well can result in improved performance. Therefore, whenever $r_t > 0$, a *retain* function is used to select which of the past executed actions and their contexts should be stored in the memory. In our experiments, the *retain* function selects the $k$ most recent actions, but other implementations are possible, as discussed in Appendix D.

## 4 A CBR POLICY AGENT TO GENERALIZE IN TEXT-BASED GAMES

Designing an agent that can act efficiently in TBGs using the described approach poses several challenges. Above all, efficient memory use is crucial to making the approach practical and scalable. Since the context selectors are used as keys for accessing values in the memory, their representation

needs to be such that contexts where similar actions were taken receive similar representations. At the same time, as the state space is exponential, context representations need to be focused only on relevant portions of the state and they need to be compressed and compact.

## 4.1 Representing the context through seeded graph attention

**State space as a knowledge graph.** Following previous works (Ammanabrolu & Riedl, 2019; Ammanabrolu & Hausknecht, 2020; Murugesan et al., 2021c), we represent the state of the game as a dynamic knowledge graph $\mathcal{G}_t = (\mathcal{V}_t, \mathcal{R}_t, \mathcal{E}_t)$, where a node $v \in \mathcal{V}_t$ represents an entity in the game, $r \in \mathcal{R}_t$ is a relation type, and an edge $v \xrightarrow{r} v' \in \mathcal{E}_t$ represents a relation of type $r \in \mathcal{R}_t$ between entities $v, v' \in \mathcal{V}_t$. In TBGs, the space of valid actions $\mathcal{A}_t$ can be modeled as a template-based action space, where actions $a_t$ are instances of a finite set of templates with a given set of entities, denoted as $\mathcal{V}_{a_t} \subseteq \mathcal{V}_t$. As an example, the action "*kill orc with sword*" can be seen as an instance of the template "*kill $v_1$ with $v_2$*", where $v_1$ and $v_2$ are "*orc*" and "*sword*" respectively.

**Seeded graph attention.** The state graph $\mathcal{G}_t$ and the entities $\mathcal{V}_{a_t}$ are provided as input to the agent for each action $a_t \in \mathcal{A}_t$, in order to build an action-specific contextualized representation of the state. A pre-trained BERT model (Devlin et al., 2019) is used to get a representation $\mathbf{h}_v^{(0)} \in \mathbb{R}^d$ for each node $v \in \mathcal{V}_t$. Inspired by Sun et al. (2018), we propose a seeded graph attention mechanism (GAT), so that the propagation of messages is weighted more for nodes close to the entities $\mathcal{V}_{a_t}$. Let $\alpha_{vu}^{(l)}$ denote the attention coefficients given by a graph attention network (Velickovic et al., 2018) at layer $l$ for nodes $v, u \in \mathcal{V}_t$. Then, for each node $v \in \mathcal{V}_t$, we introduce a coefficient $\beta_v^{(l)}$ that scales with the amount of messages received by node $v$ at layer $l$:

$$\beta_v^{(1)} = \begin{cases} \frac{1}{|\mathcal{V}_{a_t}|} & \text{if } v \in \mathcal{V}_{a_t} \\ 0 & \text{otherwise} \end{cases}, \qquad \beta_v^{(l+1)} = (1-\lambda)\beta_v^{(l)} + \lambda \sum_{u \in \mathcal{N}_v} \alpha_{vu}^{(l)} \beta_u^{(l)},$$

where $\mathcal{N}_v$ denotes the neighbors of $v$, considering the graph as undirected. Note that, at layer $l = 1$, only the nodes in $\mathcal{V}_{a_t}$ receive messages, whereas for increasing values of $l$, $\beta_v^{(l)}$ will be non-zero for their $(l-1)$-hop neighbors as well. The representation of each $v \in \mathcal{V}_t$ is then updated as:

$$\mathbf{h}_v^{(l)} = FFN^{(l)}\left(\mathbf{h}_v^{(l-1)} + \beta_v^{(l)} \sum_{u \in \mathcal{N}_v} \alpha_{vu}^{(l)} \mathbf{W}^{(l)} \mathbf{h}_u^{(l-1)}\right),$$

where $FFN^{(l+1)}$ is a 2-layer feed-forward network with ReLU non-linearity and $\mathbf{W}^{(l)} \in \mathbb{R}^{d \times d}$ are learnable parameters. Finally, we compute a final continuous contextualized representation $\mathbf{c}_{a_t}$ of the state by summing the linear projections of the hidden representations of each $v \in \mathcal{V}_{a_t}$ and passing the result through a further feed-forward network.

## 4.2 Memory access through context quantization

Given a continuous representation $\mathbf{c}_{a_t}$ of the context, we need an efficient way to access the memory $\mathcal{M}_t$ to retrieve or store actions based on such a context selector. Storing and retrieving based on the continuous representation $\mathbf{c}_{a_t}$ would be impractical for scalability reasons. Additionally, since the parameters of the agent change over the training time, the same context would result in several duplicated entries in the memory even with a pre-trained agent over different episodes.

**Discretization of the context.** To address these problems, we propose to use vector quantization (Ballard, 2000) before reading or writing to memory. Following previous work (Chen et al., 2018; Sachan, 2020), we learn a discretization function $\phi : \mathbb{R}^d \rightarrow \mathbb{Z}_K^D$, that maps the continuous representation $\mathbf{c}_{a_t}$ into a $K$-way $D$-dimensional code $c_t \in \mathbb{Z}_K^D$, with $|\mathbb{Z}_K| = K$ (we refer to $c_t$ as a KD code). With reference to Section 3, then we will use $c_t = context(s_t, a_t) = \phi(\mathbf{c}_{a_t})$ as the context selector used to access the memory $\mathcal{M}_t$. In order to implement the discretization function, we define a set of $K$ key vectors $\mathbf{k}_i \in \mathbb{R}^d, i = 1, \ldots, K$, and we divide each vector in $D$ partitions $\mathbf{k}_i^j \in \mathbb{R}^{d/D}, j = 1, \ldots, D$. Similarly, we divide $\mathbf{c}_{a_t}$ in $D$ partitions $\mathbf{c}_{a_t}^j \in \mathbb{R}^{d/D}, j = 1, \ldots, D$. Then, we compute the $j$-th code $z^j$ of $c_t$ by nearest neighbor search, as $z^j = \arg\min_i \|\mathbf{c}_{a_t}^j - \mathbf{k}_i^j\|_2^2$. We use the straight-through estimator (Bengio et al., 2013) to address the non differentialbility of the *argmin* operator.

**Memory access.** The KD codes introduced above are used to provide a memory-efficient representation of the keys in the memory. Then, given the KD code representing the current context selector $c_t$, we query the memory by computing a similarity measure $sim(c_t, c_t^{\mathcal{M}})$ between $c_t$ and each $c_t^{\mathcal{M}}$ in $\mathcal{M}_t$. The similarity function is defined as the fraction of codes shared by $c_t$ and $c_t^{\mathcal{M}}$. The context-action pair with the highest similarity is returned as a result of the memory access, together with a relevance score $\delta$ representing the value of the similarity measure.

## 4.3 SYMBOLIC ACTION REUSE AND REVISE POLICY

We use a simple purely symbolic *reuse* function to adapt the actions retrieved from the memory to the current state. Let $c_t$ be the context selector computed based on state $s_t$ and the entities $\mathcal{V}_{a_t}$, as explained in Sections 4.1 and 4.2. Denote with $(c_t^{\mathcal{M}}, a_t^{\mathcal{M}})$ the context-action pair retrieved from $\mathcal{M}_t$ with confidence $\delta$. Then, the reuse function $reuse(a_t^{\mathcal{M}}, s_t, \delta)$ constructs the action candidate $\tilde{a}_t$ as the action with the same template as $a_t^{\mathcal{M}}$ applied to the entities $\mathcal{V}_{a_t}$. If the reuse step cannot generate a valid action, we revert to the neural policy agent $\pi$ that outputs a probability distribution over the current admissible actions $\mathcal{A}_t$.

## 5 TRAINING

In Section 3, we have introduced an on-policy RL agent that relies on case-based reasoning to act in the world efficiently. This agent can be trained in principle using any online RL method. This section discusses the training strategies and learning objectives used in our implementation.

**Objective.** Two main portions of the model need to be trained: *(a)* the *retriever*, namely the neural network that computes the context representation and accesses the memory through its discretization, and *(b)* the main neural agent $\pi$ which is used in the revise step. All agents $\pi$ used in our experiments are trained with an Advantage Actor-Critic (A2C) method. For optimizing the parameters of $\pi$, we use the same learning objectives defined by Adolphs & Hofmann (2019), as described in Appendix A. Whenever the executed action $a_t^\star$ is not chosen by the model $\pi$ but it comes from the symbolic reuse step, then we optimize instead an additional objective for the retriever, namely the following contrastive loss (Hadsell et al., 2006):

$$\mathcal{L}_r^{(t)} = \frac{1}{2}(1 - y_t)(1 - sim(c_t, c_t^{\mathcal{M}}))^2 + \frac{1}{2}y_t \max\{0, \mu - 1 + sim(c_t, c_t^{\mathcal{M}})\}^2,$$

where $c_t$ denotes the context selector of the action executed at time step $t$, $c_t^{\mathcal{M}}$ is the corresponding key entry retrieved from $\mathcal{M}_t$, $\mu$ is the margin parameter of the contrastive loss, and $y_t = 1$ if $r_t > 0$, $y_t = 0$ otherwise. This objective encourages the retriever to produce similar representations for two contexts where reusing an action yielded a positive reward.

**Pretraining.** To make learning more stable and allow the agent to act more efficiently, we found it beneficial to pretrain the retriever. This minimizes large shifts in the context representations over the training time. We run a baseline agent (Ammanabrolu & Hausknecht, 2020) to collect instances of the state graph and actions that yielded positive rewards. Then we train the retriever to encode to similar representations the contexts for which similar actions (i.e., actions with the same template) were used. This is achieved using the same contrastive loss defined above.

## 6 EXPERIMENTS

This section provides a detailed evaluation of our approach. We assess quantitatively the performance of CBR combined with existing RL approaches and we demonstrate its capability to improve sample efficiency and generalize out of the training distribution. Next, we provide qualitative insights and examples of the behavior of the model and we perform an ablation study to understand the role played by the different components of the architecture.

## 6.1 EXPERIMENTAL SETUP

**Agents.** We consider several agents obtained by plugging existing RL methods in the revise step. We first define two simple approaches: **CBR-only**, where we augment a random policy with the

| | Easy | | Medium | | Hard | |
|---|---|---|---|---|---|---|
| | #Steps | Norm. Score | #Steps | Norm. Score | #Steps | Norm. Score |
| **Text** | $23.83 \pm 2.16$ | $0.88 \pm 0.04$ | $44.08 \pm 0.93$ | $0.60 \pm 0.02$ | $49.84 \pm 0.38$ | $0.30 \pm 0.02$ |
| **TPC** | $20.59 \pm 5.01$ | $0.89 \pm 0.06$ | $42.61 \pm 0.65$ | $0.62 \pm 0.03$ | $48.45 \pm 1.13$ | $0.32 \pm 0.04$ |
| **KG-A2C** | $22.10 \pm 2.91$ | $0.86 \pm 0.06$ | $41.61 \pm 0.37$ | $0.62 \pm 0.03$ | $48.00 \pm 0.61$ | $0.32 \pm 0.00$ |
| **BiKE** | $18.27 \pm 1.13$ | $0.94 \pm 0.02$ | $39.34 \pm 0.72$ | $0.64 \pm 0.02$ | $47.19 \pm 0.64$ | $0.34 \pm 0.02$ |
| **CBR-only** | $22.13 \pm 1.98$ | $0.80 \pm 0.05$ | $43.76 \pm 1.23$ | $0.62 \pm 0.03$ | $48.12 \pm 1.30$ | $0.33 \pm 0.06$ |
| **Text + CBR** | $17.53 \pm 3.36$ | $0.93 \pm 0.04$ | $39.10 \pm 1.77$ | $0.66 \pm 0.04$ | $47.11 \pm 1.21$ | $0.34 \pm 0.02$ |
| **TPC + CBR** | $16.81 \pm 3.12$ | $0.94 \pm 0.03$ | $37.05 \pm 1.61$ | $0.67 \pm 0.03$ | $47.25 \pm 1.56$ | $0.37 \pm 0.03$ |
| **KG-A2C + CBR** | $15.91 \pm 2.52$ | $0.95 \pm 0.03$ | $36.13 \pm 1.65$ | $0.66 \pm 0.05$ | $46.11 \pm 1.13$ | $0.40 \pm 0.04$ |
| **BiKE + CBR** | $15.72 \pm 1.15$ | $0.95 \pm 0.04$ | $35.24 \pm 1.22$ | $0.67 \pm 0.03$ | $45.21 \pm 0.87$ | $0.42 \pm 0.04$ |

**Table 1:** Test-set performance for *TWC in-distribution* games

| | Easy | | Medium | | Hard | |
|---|---|---|---|---|---|---|
| | #Steps | Norm. Score | #Steps | Norm. Score | #Steps | Norm. Score |
| **Text** | $29.90 \pm 2.92$ | $0.78 \pm 0.02$ | $45.90 \pm 0.22$ | $0.55 \pm 0.01$ | $50.00 \pm 0.00$ | $0.20 \pm 0.02$ |
| **TPC** | $27.74 \pm 4.46$ | $0.78 \pm 0.07$ | $44.89 \pm 1.52$ | $0.58 \pm 0.01$ | $50.00 \pm 0.00$ | $0.19 \pm 0.03$ |
| **KG-A2C** | $28.34 \pm 3.63$ | $0.80 \pm 0.07$ | $43.05 \pm 2.52$ | $0.59 \pm 0.01$ | $50.00 \pm 0.00$ | $0.21 \pm 0.00$ |
| **BiKE** | $25.59 \pm 1.92$ | $0.83 \pm 0.01$ | $41.01 \pm 1.61$ | $0.61 \pm 0.01$ | $50.00 \pm 0.00$ | $0.23 \pm 0.02$ |
| **CBR-only** | $23.43 \pm 2.09$ | $0.80 \pm 0.04$ | $44.03 \pm 1.75$ | $0.63 \pm 0.04$ | $48.71 \pm 1.15$ | $0.31 \pm 0.03$ |
| **Text + CBR** | $20.91 \pm 1.72$ | $0.89 \pm 0.02$ | $40.32 \pm 1.27$ | $0.66 \pm 0.04$ | $47.89 \pm 0.87$ | $0.32 \pm 0.06$ |
| **TPC + CBR** | $18.90 \pm 1.91$ | $0.92 \pm 0.01$ | $37.30 \pm 1.00$ | $0.66 \pm 0.02$ | $47.54 \pm 1.67$ | $0.34 \pm 0.03$ |
| **KG-A2C + CBR** | $18.21 \pm 1.32$ | $0.90 \pm 0.02$ | $37.02 \pm 1.22$ | $0.68 \pm 0.03$ | $47.10 \pm 1.12$ | $0.38 \pm 0.02$ |
| **BiKE + CBR** | $17.15 \pm 1.45$ | $0.93 \pm 0.03$ | $35.45 \pm 1.40$ | $0.67 \pm 0.03$ | $45.91 \pm 1.32$ | $0.40 \pm 0.03$ |

**Table 2:** Test-set performance for *TWC out-of-distribution* games

CBR approach, and **Text + CBR**, which relies on the CBR method combined with a simple GRU-based policy network that consumes as input the textual observation from the game. Next, we select three recently proposed TBG approaches: **Text+Commonsense** (**TPC**) (Murugesan et al., 2021b), **KG-A2C** (Ammanabrolu & Hausknecht, 2020), and **BiKE** (Murugesan et al., 2021c), to create the **TPC + CBR**, **KG-A2C + CBR** and **BiKE + CBR** agents. We treat the original agents as baselines.

**Datasets.** We empirically verify the efficacy of our approach on **TextWorld Commonsense** (*TWC*) (Murugesan et al., 2021b) and **Jericho** (Hausknecht et al., 2020). *Jericho* is a well-known and challenging learning environment including 33 interactive fiction games. *TWC* is an environment which builds on *TextWorld* (Côté et al., 2018) and provides a suite of games requiring commonsense knowledge. *TWC* allows agents to be tested on two settings: the *in-distribution games*, where the objects that the agent encounters in the test set are the same as the objects in the training set, and the *out-of-distribution games* which have no entity in common with the training set. For each of these settings, *TWC* provides three difficulty levels: *easy*, *medium*, and *hard*.

**Evaluation metrics.** Following Murugesan et al. (2021b), we evaluate the agents on *TWC* based on the number of steps (**#Steps**) required to achieve the goal (lower is better) and the normalized cumulative reward (**Norm. Score**) obtained by the agent (larger is better). On *Jericho*, we follow previous work (Hausknecht et al., 2020; Guo et al., 2020; Ammanabrolu & Hausknecht, 2020) and we report the average score achieved over the last 100 training episodes.

## 6.2 RESULTS ON TEXTWORLD COMMONSENSE

Table 1 reports the results on *TWC* for the *in-distribution* set of games. Overall, we observe that CBR consistently improves the performance of all the baselines. The performance boost is large enough that even a simple method as **Text + CBR** outperforms all considered baselines except **BiKE**.

**Out-of-distribution generalization.** CBR's ability to retrieve similar cases should allow our method to better generalize to new and unseen problems. We test this hypothesis on the *out-of-*

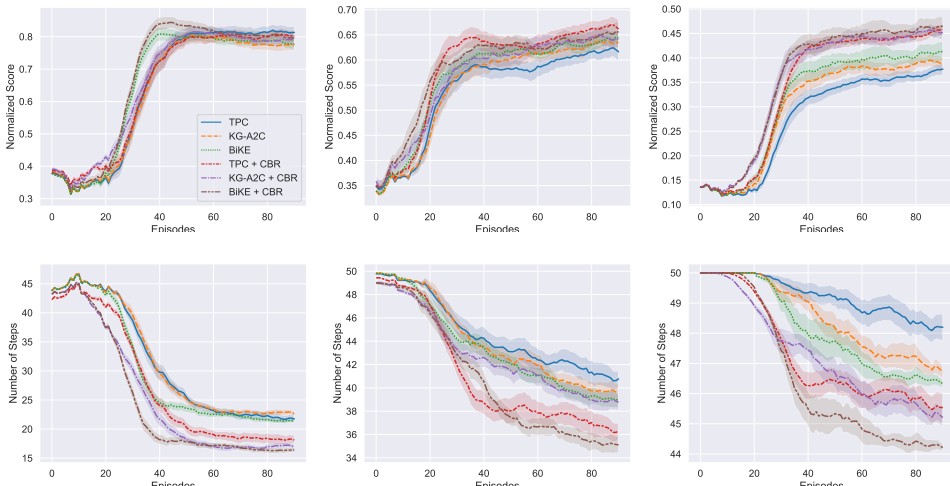

**Figure 2:** Performance on *TWC* (showing mean and standard deviation averaged over 5 runs) for the three difficulty levels: *easy* (left), *medium* (middle), *Hard* (right) using normalized score and number of steps.

*distribution* games in *TWC*. The results of this experiment are reported in Table 2. We notice that all existing approaches fail to generalize out of the training distribution and suffer a substantial drop in performance in this setting. However, when coupled with CBR, the drop is minor (on average **6%** with CBR *vs* **35%** without on the hard level). Interestingly, even the **CBR-only** agent achieves competitive results compared to the top-performing baselines.

**Sample efficiency.**    Another key benefit of our approach comes as better sample efficiency. With its ability to explicitly store prior solutions effectively, CBR allows existing algorithms to learn faster. Figure 2 shows the learning curves for our best agents and the corresponding baselines. The plots report the performance of the agent over the training episodes, both in terms of the number of steps and the normalized score. Overall, we observe that the CBR agents obtain faster convergence to their counterparts on all difficulty levels.

### 6.3    Performance on the Jericho games

We evaluate our best performing variant from the experiments on *TWC* (**BiKE + CBR**) against existing approaches on the 33 games in the *Jericho* environment. We compare our approach against strong baselines, including **TDQN** (Hausknecht et al., 2020), **DRRN** (He et al., 2016), **KG-A2C** (Ammanabrolu & Hausknecht, 2020), **MPRC-DQN** (Guo et al., 2020), and **RC-DQN** (Guo et al., 2020). The same experimental setting and handicaps as the baselines are used, as we train for 100 000 steps and we assume access to valid actions. Table 3 summarizes the results of the *Jericho* games. We observe that our CBR agent achieves comparable or better performance than any baseline on 24 (73%) of the games, strictly outperforming all the other agents in 18 games.

### 6.4    Qualitative analysis and ablation studies

**Insights on the model.**    Figure 3 provides two examples showing the **BiKE + CBR** agent interacting with the zork1 game. In the example on top, the agent retrieves an experience that can be successfully reused and turned into a valid action at the current time step. The heat maps visualize the value of the context similarity function defined in Section 4 for the top entries in the memory. In the negative example at the bottom instead, the agent retrieves an action that is not useful and needs to fall back to the neural policy $\pi$. Figure 4 (top) shows the fraction of times that actions retrieved from the memory are reused successfully in the *TWC* games. We observe that, both for *in-distribution* and *out-of-distribution* games, the trained agent relies on CBR from 60% to approximately 70% of the times. Figure 4 (bottom) further shows the fraction of times that the neural agent would have been able to select a rewarded action as well, when the CBR reuses a successful action. The plot shows that, for the *out-of-distribution* games, the neural agent would struggle to select good actions when the CBR is used.

| Game | Human (max) | Human (Walkthrough-100) | TDQN | DRRN | KG-A2C | MPRC-DQN | RC-DQN | BiKE + CBR |
|------|------|------|------|------|------|------|------|------|
| 905 | 1 | 1 | **0** | **0** | **0** | **0** | **0** | **0** |
| acorncourt | 30 | 30 | 1.6 | 10 | 0.3 | 10 | 10 | **12.2** |
| adventureland | 100 | 42 | 0 | 20.6 | 0 | 24.2 | 21.7 | **27.3** |
| afflicted | 75 | 75 | 1.3 | 2.6 | – | **8** | **8** | 3.2 |
| awaken | 50 | 50 | **0** | **0** | **0** | **0** | **0** | **0** |
| detective | 360 | 350 | 169 | 197.8 | 207.9 | 317.7 | 291.3 | **326.1** |
| dragon | 25 | 25 | -5.3 | -3.5 | 0 | 0.04 | 4.84 | **8.3** |
| inhumane | 90 | 70 | 0.7 | 0 | 3 | 0 | 0 | **24.2** |
| library | 30 | 30 | 6.3 | 17 | 14.3 | 17.7 | 18.1 | **22.3** |
| moonlit | 1 | 1 | **0** | **0** | **0** | **0** | **0** | **0** |
| omniquest | 50 | 50 | 16.8 | 10 | 3 | 10 | 10 | **17.2** |
| pentari | 70 | 60 | 17.4 | 27.2 | 50.7 | 44.4 | 43.8 | **52.1** |
| reverb | 50 | 50 | 0.3 | **8.2** | – | 2 | 2 | 6.5 |
| snacktime | 50 | 50 | 9.7 | 0 | 0 | 0 | 0 | **22.1** |
| temple | 35 | 20 | 7.9 | 7.4 | 7.6 | **8** | **8** | 7.8 |
| ztuu | 100 | 100 | 4.9 | 21.6 | 9.2 | 85.4 | 79.1 | **87.2** |
| advent | 350 | 113 | 36 | 36 | 36 | **63.9** | 36 | 62.1 |
| balances | 51 | 30 | 4.8 | 10 | 10 | 10 | 10 | **11.9** |
| deephome | 300 | 83 | **1** | **1** | **1** | **1** | **1** | **1** |
| gold | 100 | 30 | **4.1** | 0 | – | 0 | 0 | 2.1 |
| jewel | 90 | 24 | 0 | 1.6 | 1.8 | 4.46 | 2 | **6.4** |
| karn | 170 | 40 | 0.7 | 2.1 | 0 | **10** | **10** | 0 |
| ludicorp | 150 | 37 | 6 | 13.8 | 17.8 | 19.7 | 17 | **23.8** |
| yomomma | 35 | 34 | 0 | 0.4 | – | **1** | **1** | **1** |
| zenon | 20 | 20 | 0 | 0 | 3.9 | 0 | 0 | **4.1** |
| zork1 | 350 | 102 | 9.9 | 32.6 | 34 | 38.3 | 38.8 | **44.3** |
| zork3 | 7 | 3 | 0 | 0.5 | 0.1 | **3.63** | 2.83 | 3.2 |
| anchor | 100 | 11 | **0** | **0** | **0** | **0** | **0** | **0** |
| enchanter | 400 | 125 | 8.6 | 20 | 12.1 | 20 | 20 | **36.3** |
| sorcerer | 400 | 150 | 5 | 20.8 | 5.8 | **38.6** | 38.3 | 24.5 |
| spellbrkr | 600 | 160 | 18.7 | 37.8 | 21.3 | 25 | 25 | **41.2** |
| spirit | 250 | 8 | 0.6 | 0.8 | 1.3 | 3.8 | **5.2** | 4.2 |
| tryst205 | 350 | 50 | 0 | 9.6 | – | 10 | 10 | **13.4** |
| **Best agent** | | | 6 (18%) | 6 (18%) | 5 (15%) | 12 (36%) | 10 (30%) | 24 (73%) |

**Table 3:** Average raw score on the *Jericho* games. We denote with colors the difficulty of the games (green for *possible* games, yellow for *difficult* games and red for *extreme* games). The last row reports the fraction and the absolute number of games where an agent achieves the best score. We additionally report human performance (**Human – max**) and the 100-step results from a human-written walkthrough (**Human – Walkthrough 100**). Results are taken from the original papers or "−" is used if a result was not reported.

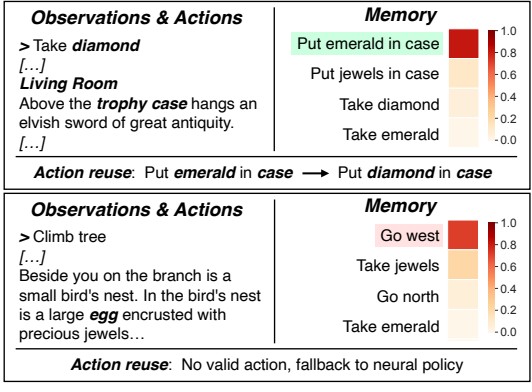

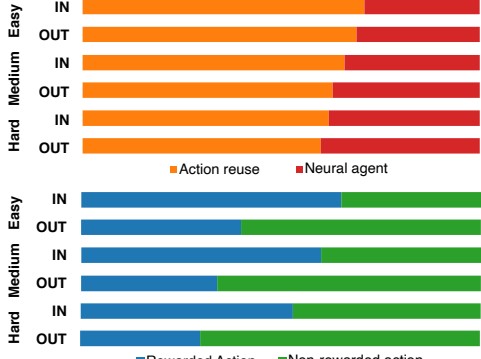

**Figure 3:** Examples from the zork1 game, showing the content of the memory and the context similarities, in a situation where the agent is able to reuse a previous experience and in a case where the revise step is needed.

**Figure 4:** Fraction of times that a retrieved action is reused successfully on *TWC* (top). Fraction of times that the neural agent would have picked a rewarded action when CBR is used successfully (bottom).

**Ablation studies.** In order to understand the role of the main modules of our CBR agent, we designed some ablation studies. First, instead of using the seeded GAT, we define the context of a state-action pair $context(s_t, a_t)$ as just one of the entities that $a_t$ is applied to. This definition suits well the *TWC* games because rewarded actions are always applied to one target object and a location for that object (see Appendix G for details). Note that, since the set of entities is discrete, no context quantization is needed. We report the performance of the resulting **BiKE + CBR (w/o GAT)** agent in

|    |                        | Easy            | Medium          | Hard            |
|----|------------------------|-----------------|-----------------|-----------------|
| IN | BiKE + CBR (w/o GAT)   | 16.32 ± 1.10    | 36.13 ± 1.40    | 45.72 ± 0.63    |
|    | BiKE + CBR (w/o VQ)    | 22.67 ± 1.23    | 43.18 ± 2.10    | 49.21 ± 0.55    |
| OUT| BiKE + CBR (w/o GAT)   | 18.15 ± 1.51    | 37.10 ± 1.41    | 46.70 ± 0.71    |
|    | BiKE + CBR (w/o VQ)    | 27.75 ± 2.11    | 44.55 ± 1.67    | 50.00 ± 0.00    |

**Table 4:** Results of the ablation study on *TWC*, evaluated based on the number of steps (**#Steps**) to solve the games.

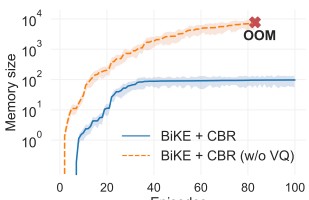

**Figure 5:** Number of entries in the memory over training.

Table 4. The results show that CBR on *TWC* is effective even with this simple context definition, but the lower performance of the agent demonstrates the advantage of incorporating additional context information. Finally, we investigate the role played by vector quantization, by experimenting with an agent (**BiKE + CBR w/o VQ**) that stores the continuous context representations. In general, this poses scalability challenges, but since *TWC* has only 5 games per difficulty level, each with a small number of objects, we were able to evaluate the performance of this agent on the three levels separately. The results, reported in Table 4, show that this agent performs much worse than the other CBR implementations. This happens because storing continuous representations over the training results in duplicated entries in the memory and makes it harder to retrieve meaningful experiences. Figure 5 demonstrates how the size (number of entries) in the memory grows over the training time. In this experiment, we trained the agent on all difficulty levels at the same time, resulting in the implementation running out of memory (OOM) on the GPU. More ablation studies are reported in Appendix C, D, E, F, and G.

## 7 RELATED WORK

**Text-based RL.** TBGs are a rich domain for studying grounded language understanding and how text information can be utilized in control. Prior work has explored text-based RL to learn strategies for multi-user dungeon games (Narasimhan et al., 2015) and other environments (Branavan et al., 2012). Zahavy et al. (2018) proposed the Action-Elimination Deep Q-Network (AE-DQN), which learns to predict invalid actions in the text-adventure game *Zork*. Recently, Côté et al. (2018) introduced `TextWorld`, a sandbox learning environment for training and evaluating RL agents on text-based games. On the same line, Murugesan et al. (2021b) introduced *TWC* that requires agents with commonsense knowledge (Murugesan et al., 2020; Basu et al., 2021). The *LeDeepChef* system (Adolphs & Hofmann, 2019) achieved good results on the *First TextWord Problems* (Trischler et al., 2019) by supervising the model with entities from `FreeBase`, allowing the agent to generalize to unseen objects. A recent line of work learns symbolic (typically graph-structured) representations of the agent's belief. Notably, Ammanabrolu & Riedl (2019) proposed *KG-DQN* and Adhikari et al. (2020) proposed *GATA*. We also use graphs to model the state of the game.

**Case-based reasoning in RL.** In the context of RL, CBR has been used to speed up and improve transfer learning in heuristic-based RL. Celiberto Jr et al. (2011) and Bianchi et al. (2018) have shown that cases collected from one domain can be used as heuristics to achieve faster convergence when learning an RL algorithm on a different domain. In contrast to these works, we present a scalable way of using CBR alongside deep RL methods in settings with very large state spaces. More recently, CBR has been successfully applied in the field of knowledge-based reasoning. Das et al. (2020) and Das et al. (2021) show that CBR can effectively learn to generate new logical reasoning chains from prior cases, to answer questions on knowledge graphs.

## 8 CONCLUSION AND FUTURE WORK

In this work, we proposed new agents for TBGs using case-based reasoning. In contrast to expensive deep RL approaches, CBR simply builds a collection of its past experiences and uses the ones relevant to the current situation to decide upon its next action in the game. Our experiments showed that CBR when combined with existing RL agents can make them more efficient and aid generalization in out-of-distribution settings. Even though CBR was quite successful in the TBGs explored in our work, future work is needed to understand the limitations of CBR in such settings.

## ACKNOWLEDGEMENTS

This work was funded in part by an IBM fellowship to SD and in part by a small project grant to MS from the Hasler Foundation.

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

## A    TRAINING DETAILS

All agents used in our experiments are trained with an Advantage Actor Critic (A2C) method according to the general scheme defined below. We use the $n$-step temporal difference method (Sutton & Barto, 2018) to compute the return $R(s_t, a_t)$ of a single time step $t$ in a session of length $T$ as:

$$R(s_t, a_t) = \gamma^{T-t} V(s_T) + \sum_{i=0}^{T-t} \gamma^i r(s_{t+i}, a_{t+i}),$$

where $V(s_T)$ denotes the value of $s_T$ computed by the critic network, $\gamma$ is the discount factor, and $r$ is the reward function. We then compute the advantage $A(s_t, a_t) = R(s_t, a_t) - V(s_t)$. The final objective term consists of four separate objectives. First, we have $\mathcal{L}_\pi^{(t)}$ that denotes the objective of the policy, which tries to maximize the the advantage $A$:

$$\mathcal{L}_\pi^{(t)} = -A(s_t, a_t^\star) \log \pi(a_t^\star | s_t).$$

We then add the objective of the critic as:

$$\mathcal{L}_v^{(t)} = \frac{1}{2} \left( R(s_t, a_t^\star) - V(s_t) \right)^2.$$

$\mathcal{L}_v^{(t)}$ encourages the value of the critic $V$ to better estimate the reward $R$ by reducing the mean squared error between them. To prevent the policy from assigning a large weight on a single action, we perform entropy regularization by introducing an additional term:

$$\mathcal{L}_e^{(t)} = \eta \cdot \sum_{a_t \in \mathcal{A}_t} \pi(a_t | s_t) \cdot \log \pi(a_t | s_t).$$

The $\eta$ parameter helps balance the exploration-exploitation trade-off for the policy. The sum of the above objectives defines the loss when the neural agent $\pi$ is used to select the action $a_t^\star$. In case actions are reused from the memory, then we use the contrastive loss as discussed in Section 5, namely we compute the loss as:

$$\mathcal{L}_r^{(t)} = \frac{1}{2}(1 - y_t)(1 - sim(c_t, c_t^{\mathcal{M}}))^2 + \frac{1}{2} y_t \max\{0, \mu - 1 + sim(c_t, c_t^{\mathcal{M}})\}^2.$$

## B    ENHANCING BASELINE AGENTS WITH CBR ON JERICHO

In this section, we report additional experimental results on a subset of the *Jericho* games, in order to show the performance improvement obtained by different baseline agents when enhanced with case-based reasoning. Table 5 shows the results obtained when coupling the agents described in Section 6.1 with CBR. Similarly to what we discussed for *TWC* in Section 6.2, we observe that CBR consistently improves the performance of all the agents. The best performing agent is **BiKE + CBR**, which is the agent that we evaluated on the complete set of games in Section 6.3.

| Game | KG-A2C | KG-A2C + CBR | Text | Text + CBR | TPC | TPC + CBR | BiKE | BiKE + CBR |
|---|---|---|---|---|---|---|---|---|
| detective | 207.9 | **255.6** | 205.8 | **242.3** | 245.6 | **315.1** | 278.2 | **326.1** |
| inhumane | 3 | **15.6** | 1.1 | **14.5** | 4.5 | **18.3** | 9.2 | **24.2** |
| snacktime | 0 | **15.5** | 8.1 | **9.8** | 15.7 | **19.3** | 18.8 | **22.1** |
| karn | **0** | **0** | **0** | **0** | **0** | **0** | **0** | **0** |
| zork1 | 34 | **34.2** | 31.5 | **38.2** | 36 | **39.2** | 39.5 | **44.3** |
| zork3 | 0.1 | **1.7** | 0 | **1.6** | 1.7 | **1.9** | 2.5 | **3.2** |
| enchanter | 12.1 | **26.2** | 10.1 | **26.4** | 13.2 | **24.5** | 19.7 | **36.3** |
| spellbrkr | 21.3 | **36.1** | 20.4 | **33.2** | 38.1 | **40.2** | 38.8 | **41.2** |

**Table 5:** Additional results on *Jericho* showing the performance improvement obtained by enhancing several baseline agents with CBR.

## C    ABLATION STUDY ON MEMORY ACCESS

In this section, we investigate the effectiveness of our approach based on vector quantization for efficient memory access. We consider several variants, where VQ is either dropped completely or replaced with other techniques.

## C.1 Alternatives to vector quantization

All the agents we consider are variants of the best-performing agent on *Jericho* and *TWC*, namely the **BiKE + CBR** agent. We experiment with the following techniques.

- **BiKE + CBR (w/o VQ)** completely removes the vector quantization and stores the continuous context representations as keys in the case memory. In order to make this feasible, for the experiments on *Jericho*, we limit the size of the memory to the 5000 most recent entries.
- **BiKE + CBR** (**RP**) relies on random projection (RP) in order to reduce the dimensionality of the context representations stored by the CBR approach. In this case, each context representation is projected into a $p$-dimensional space using a random matrix $R \in \mathbb{R}^{p \times d}$, with components drawn from a normal distribution $N(0, \frac{1}{p})$ with mean 0 and standard deviation $\frac{1}{p}$.
- **BiKE + CBR (SRP)** employs sign random projection (SRP) in order to obtain a discrete representation of the context. In this variant, after projection to a $p$-dimensional space, the context representations are discretized by applying an element-wise sign function.
- **BiKE + CBR (LSH)** replaces vector quantization with locality sensitive hashing (LSH). In this case, context representations are converted into $h$-bit hash codes for $l$ different hash tables. The retrieve step selects the representation with the highest cosine similarity to the query context encoding, among the vectors falling in the same bucket.

## C.2 Results and discussion

Table 6 shows the scores achieved by each variant of the **BiKE + CBR** method on a subset of the *Jericho* games. We notice that the approaches relying on continuous context representations (**w/o VQ** and **RP**) perform poorly compared to the others and to the plain **BiKE** agent. This confirms our hypothesis that CBR needs discrete context representations to make the retrieve step more stable.

The **BiKE + CBR (LSH)** agent, which relies on locality sensitive hashing, achieves competitive results and consistently outperforms the **BiKE** agent. This shows that locality sensitive hashing can be a viable alternative to vector quantization. However, we observe that our **BiKE + CBR** agent, based on vector quantization, achieves better results on almost all the games, confirming the benefit of learning the discrete representation as well. Note also that LSH requires storing the complete continuous representations to be able to detect false positives, namely contexts with the same hash code as the query vector, but low cosine similarity. This makes this alternative less memory efficient compared to our implementation based on VQ.

| Game | BiKE + CBR (w/o VQ) | BiKE + CBR (RP) | BiKE + CBR (SRP) | BiKE + CBR (LSH) | BiKE + CBR |
|---|---|---|---|---|---|
| detective | 205.2 | 203.1 | 223.2 | 319.1 | **326.1** |
| inhumane | 1.5 | 3.2 | 1.1 | 20.1 | **24.2** |
| snacktime | 9.1 | 14.3 | 13.2 | 20.3 | **22.1** |
| karn | **0** | **0** | **0** | **0** | **0** |
| zork1 | 31.5 | 36.3 | 40.5 | 41.2 | **44.3** |
| zork3 | 0 | 2.7 | 2.7 | **3.6** | 3.2 |
| enchanter | 10.2 | 9.2 | 10.6 | 35.3 | **36.3** |
| spellbrkr | 21.3 | 23.8 | 30.8 | 39.3 | **41.2** |

**Table 6:** Ablation study showing the results obtained on a subset of the *Jericho* games when the vector quantization is removed (**w/o VQ**) or replaced with random projection (**RP**), sign random projection (**SRP**) or locality sensitive hashing (**LSH**).

# D Ablation study on the retain module

In this section, we evaluate different alternatives to select which actions should be retained in the memory of the agent.

## D.1 Alternatives for the retain module

In our main experiments described in Section 6, the retain module has been implemented to store the last $k$ context-action pairs in the memory, whenever the reward obtained by the agent is positive.

For *Jericho*, we set $k = 3$, as specified in Appendix J. However, other design choices are possible. We consider the following variants of the **BiKE + CBR** agent.

- **BiKE + CBR (rewarded action only)** retains only the rewarded context-action pair, without sampling any of the previous actions. This variant is a simple implementation that works well in practice, but may fail to identify useful actions that were not rewarded.

- **BiKE + CBR (TD error)** samples previous context-action pairs based on the temporal difference (TD) error. In details, the agent still retains $k$ context-action pairs: whenever the reward $r_t$ at time step $t$ is positive, the rewarded context-action pair $(c_t^\star, a_t^\star)$ is retained, together with $k - 1$ additional pairs sampled from the current trajectory, with a probability proportional to the TD error.

## D.2 RESULTS AND DISCUSSION

Table 7 shows the scores obtained by the agents on a subset of the *Jericho* games. Our main implementation storing the last $k = 3$ actions achieves the best results, whereas the **BiKE + CBR (rewarded action only)** agent performs slightly worse than the other two implementations. The agent based on the TD error achieves competitive results and a state-of-the-art score on the **snacktime** game. We observe that all variants perform well in practice and achieve overall better results than the baselines in Table 3.

| Game | BiKE + CBR (rewarded action only) | BiKE + CBR (TD error) | BiKE + CBR |
|---|---|---|---|
| detective | 324.1 | 324.8 | **326.1** |
| inhumane | 20.1 | 23.5 | **24.2** |
| snacktime | 20.3 | **23.4** | 22.1 |
| karn | **0** | **0** | **0** |
| zork1 | 40.2 | 42.4 | **44.3** |
| zork3 | **3.2** | **3.2** | **3.2** |
| enchanter | 35.3 | 34.1 | **36.3** |
| spellbrkr | 40.3 | 40.8 | **41.2** |

**Table 7:** Ablation study showing the results obtained on a subset of the *Jericho* games when only the context-action pair achieving positive reward is retained (**rewarded action only**), when context-action pairs are sampled using the TD error (**TD error**), and when the last 3 pairs are retained (**BiKE + CBR**).

## E    TRAINING THE AGENT AND THE RETRIEVER JOINTLY

In this experiment, we try to understand if training the neural agent and the CBR retriever jointly would improve upon our choice of just training the two networks separately.

Our implementation works by training the neural agent $\pi$ and the CBR retriever separately with different objectives, as described in Appendix A. However, we can make the neural agent aware of the CBR retriever and train the two networks jointly. In this case, we need to modify the architecture of the neural agent $\pi$ in order to take into account the action candidates $\tilde{\mathcal{A}}_t$ produced by the CBR. In details, we compute an action selector by attention between the valid actions $\mathcal{A}_t$ and the candidates $\tilde{\mathcal{A}}_t$ and we concatenate this action selector to the vector used by the neural agent to score admissible actions. Then, the agent and the retriever can be trained jointly, optimizing an objective given by the sum of all the losses in Section A.

Table 8 and 9 show the results obtained by the baseline agents when they are trained jointly with the CBR retriever. On *TWC*, we observe that the joint variants of the agents achieve comparable results with their counterparts in Table 1 and 2. On *Jericho*, the agents trained jointly with the retriever achieve strong results, but they perform slightly worse than our main approach that keeps the retriever separate from the neural agent. This shows that the joint version is slightly harder to train. Also, it brings the disadvantage that the architecture of the neural agent has to be changed to take the CBR into account, whereas our main approach that keeps the retriever separate allows readily plugging any on-policy agent for TBGs.

| | | Easy | | Medium | | Hard | |
|---|---|---|---|---|---|---|---|
| | | #Steps | Norm. Score | #Steps | Norm. Score | #Steps | Norm. Score |
| **IN** | **Text + CBR (joint)** | $17.91 \pm 3.80$ | $0.91 \pm 0.04$ | $40.22 \pm 1.70$ | $0.65 \pm 0.05$ | $47.94 \pm 1.10$ | $0.33 \pm 0.02$ |
| | **TPC + CBR (joint)** | $16.80 \pm 1.97$ | $0.94 \pm 0.04$ | $36.12 \pm 1.32$ | $0.67 \pm 0.03$ | $46.10 \pm 0.72$ | $0.41 \pm 0.03$ |
| | **KG-A2C + CBR (joint)** | $16.40 \pm 1.89$ | $0.95 \pm 0.05$ | $36.50 \pm 1.13$ | $0.68 \pm 0.03$ | $46.58 \pm 0.91$ | $0.40 \pm 0.06$ |
| | **BiKE + CBR (joint)** | $16.01 \pm 1.37$ | $0.94 \pm 0.03$ | $35.93 \pm 1.11$ | $0.67 \pm 0.05$ | $46.11 \pm 1.14$ | $0.42 \pm 0.04$ |
| **OUT** | **Text + CBR (joint)** | $21.47 \pm 2.32$ | $0.88 \pm 0.07$ | $39.10 \pm 1.33$ | $0.67 \pm 0.02$ | $48.10 \pm 0.92$ | $0.31 \pm 0.03$ |
| | **TPC + CBR (joint)** | $17.89 \pm 1.82$ | $0.93 \pm 0.01$ | $38.11 \pm 1.33$ | $0.65 \pm 0.03$ | $47.92 \pm 1.55$ | $0.34 \pm 0.03$ |
| | **KG-A2C + CBR (joint)** | $18.19 \pm 2.12$ | $0.93 \pm 0.02$ | $37.72 \pm 2.91$ | $0.66 \pm 0.03$ | $47.53 \pm 1.11$ | $0.40 \pm 0.04$ |
| | **BiKE + CBR (joint)** | $18.12 \pm 1.21$ | $0.94 \pm 0.05$ | $35.77 \pm 1.05$ | $0.69 \pm 0.05$ | $46.16 \pm 1.00$ | $0.40 \pm 0.04$ |

**Table 8:** Test-set results obtained on *TWC in-distribution* (**IN**) and *out-of-distribution* (**OUT**) games training different neural agents and the CBR retriever jointly.

| Game | KG-A2C + CBR (joint) | Text + CBR (joint) | TPC + CBR (joint) | BiKE + CBR (joint) |
|---|---|---|---|---|
| **detective** | 250.1 | 241.5 | **316.2** | 324.2 |
| **inhumane** | 15.1 | 13.2 | 16.3 | 24.1 |
| **snacktime** | 13.2 | **11.2** | **20.1** | 21.8 |
| **karn** | 0 | 0 | 0 | 0 |
| **zork1** | **36.4** | 36.5 | 36.3 | 43.7 |
| **zork3** | 1.5 | 1.5 | 1.8 | **3.6** |
| **enchanter** | 25.1 | 26.1 | 21.1 | 35.6 |
| **spellbrkr** | 32.3 | **33.3** | 39.2 | **41.8** |

**Table 9:** Results obtained on a subset of the *Jericho* games training different neural agents and the CBR retriever jointly. Bold values indicate when the joint variant achieves better scores than the main counterparts reported in Table 5.

## F MULTI-PARAGRAPH TEXT-BASED RETRIEVER

Knowledge graphs have been used extensively in text-based games and other areas of natural language understanding (Ammanabrolu & Hausknecht, 2020; Atzeni & Atzori, 2018; Kapanipathi et al., 2020). This section describes an alternative to our graph-based retriever, which only relies on the textual observations without modeling the state of the game as a graph.

Recent work (Guo et al., 2020) has shown that enriching the current observation with relevant observations retrieved from the history of interactions with the environment can achieve competitive results on *Jericho*. Therefore, in order to assess the effectiveness of our graph-based implementation, we compare to a multi-paragraph text-based retriever (MTPR) inspired by the work of Guo et al. (2020). In this case, we do not model the state as a graph and, subsequently, we remove the seeded graph attention mechanism from the retriever. Instead, given the current natural language observation $o_t$, we compute an action-specific representation following Guo et al. (2020), concatenating $o_t$ with the $n$ most recent observations that share objects with it or with the given action. The encoded observation is then discretized using vector quantization as in our main architecture.

We evaluated the text-based retriever on *Jericho*, integrating it in the same baseline agents described in Section 6.1. Table 10 shows the scores obtained by the agents. We observe that, overall, the graph-based retriever performs better on the vast majority of the games. This result confirms the ability of our approach based on seeded graph attention to extract relevant information from the state of the game, compared to a retriever that only relies on text information.

## G ADDITIONAL RESULTS USING ENTITIES AS CONTEXT SELECTORS

As mentioned in Section 6.4, we performed an ablation study where the seeded graph attention and the state graph where not used in the retriever. Instead, we represent the context as just a single focus entity. This choice suits very well the *TWC* games, as the goal of each game is to tidy up a house by putting objects in their commonsensical locations. Hence, each rewarded action in *TWC* is of the form "*put o on s*" or "*insert o in c*", where $o$ is an entity of type *object*, $s$ is a *supporter*, and $c$ is a

| Game | KG-A2C + CBR (MPTR) | Text + CBR (MPTR) | TPC + CBR (MPTR) | BiKE + CBR (MPTR) |
|---|---|---|---|---|
| detective | 245.2 | 233.7 | 302.3 | 321.2 |
| inhumane | 13.4 | 13.2 | 12.3 | 20.3 |
| snacktime | 12.1 | 8.2 | 18.1 | 19.5 |
| karn | 0 | 0 | 0 | 0 |
| zork1 | **36.2** | 36.2 | 37.4 | 42.2 |
| zork3 | 0.8 | 1.2 | **3.2** | **3.6** |
| enchanter | 19.3 | 24.3 | 20.2 | 32.1 |
| spellbrkr | 31.3 | 30.3 | 39.3 | 40.8 |

**Table 10:** Results obtained on a subset of the *Jericho* games using the multi-paragraph text-based retriever (**MPTR**) instead of the graph-based one. Bold values indicate when the **MPTR** variant achieves better scores than the main counterparts reported in Table 5.

*container* (Murugesan et al., 2021c; Côté et al., 2018). As an example, a rewarded action could be "*insert dirty singlet in washing machine*", where *dirty singlet* is the *object* and *washing machine* is the container.

Representing the context as just single entities of type *object*, means that the memory of the CBR agent is storing what action to apply to each object in the game, and therefore the agent is in practice constructing a registry where each object is paired with its commonsensical location. Let $c_v, c_u$ be two context entities. The *retriever* then computes the similarity between the contexts as:

$$sim(c_v, c_u) = cosine(FFN(\mathbf{h}_v), FFN(\mathbf{h}_u)),$$

where *cosine* denotes the cosine similarity, *FFN* is a 2-layer feed-forward network and $\mathbf{h}_v, \mathbf{h}_u$ are the BERT encodings of the [CLS] token for objects $v$ and $u$ respectively. The retriever is therefore encouraged to map objects that should be placed in the same location (either a *supporter* or a *container*) to similar representations.

Figure 6 depicts a $t$-SNE (van der Maaten & Hinton, 2008) visualization of the representations $FFN(\mathbf{h}_v)$ learned by the retriever of the **BiKE + CBR (w/o GAT)** agent. The plot shows that entities that belong to the same location are mapped to similar representations. This holds both for objects in the *in-distribution* games and for objects in the *out-of-distribution* games.

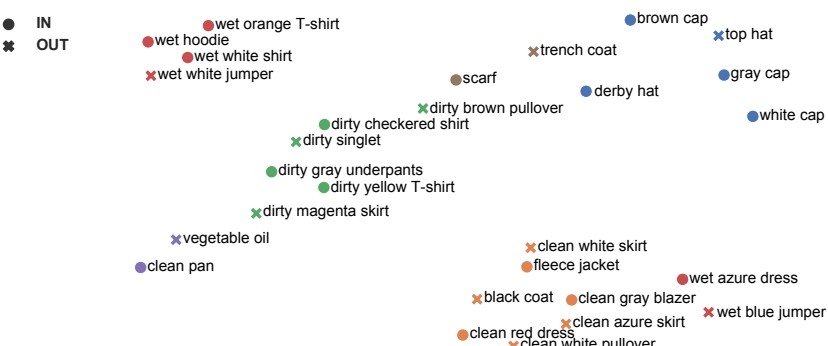

**Figure 6:** Visualization of the entity representations learned by the retriever. Colors denote the target location of each object.

We evaluated all the CBR agents with the simple retriever defined above. Table 11 reports the results for the *in-distribution* and *out-of-distribution* games. The results confirm that the entity-based context selection performs well and achieves good out-of-distribution generalization. However, we remark that the complete retriever described in Section 4 consistently achieves better results, showing the importance of incorporating additional structured information.

|  |  | Easy | | Medium | | Hard | |
|---|---|---|---|---|---|---|---|
|  |  | #Steps | Norm. Score | #Steps | Norm. Score | #Steps | Norm. Score |
| **IN** | CBR (w/o GAT) | 22.70 ± 2.05 | 0.81 ± 0.07 | 44.13 ± 1.15 | 0.62 ± 0.04 | 48.05 ± 1.30 | 0.33 ± 0.05 |
|  | Text + CBR (w/o GAT) | 19.01 ± 3.99 | 0.89 ± 0.05 | 40.10 ± 1.52 | 0.67 ± 0.05 | 47.80 ± 1.32 | 0.33 ± 0.02 |
|  | TPC + CBR (w/o GAT) | 17.15 ± 2.91 | 0.94 ± 0.04 | 38.32 ± 1.76 | 0.66 ± 0.03 | 47.22 ± 1.35 | 0.36 ± 0.03 |
|  | KG-A2C + CBR (w/o GAT) | 16.67 ± 2.30 | 0.96 ± 0.03 | 38.05 ± 1.84 | 0.66 ± 0.04 | 46.45 ± 1.02 | 0.38 ± 0.02 |
|  | BiKE + CBR (w/o GAT) | 16.32 ± 1.10 | 0.95 ± 0.03 | 36.13 ± 1.40 | 0.67 ± 0.04 | 45.72 ± 0.63 | 0.41 ± 0.03 |
| **OUT** | CBR (w/o GAT) | 23.90 ± 2.17 | 0.79 ± 0.05 | 44.71 ± 1.50 | 0.61 ± 0.04 | 48.87 ± 1.89 | 0.31 ± 0.03 |
|  | Text + CBR (w/o GAT) | 21.64 ± 2.52 | 0.88 ± 0.02 | 41.12 ± 1.21 | 0.66 ± 0.05 | 48.00 ± 1.10 | 0.32 ± 0.06 |
|  | TPC + CBR (w/o GAT) | 19.82 ± 2.13 | 0.92 ± 0.03 | 39.34 ± 1.01 | 0.67 ± 0.02 | 47.33 ± 1.30 | 0.36 ± 0.04 |
|  | KG-A2C + CBR (w/o GAT) | 19.07 ± 2.50 | 0.92 ± 0.02 | 38.41 ± 1.94 | 0.65 ± 0.04 | 46.89 ± 2.21 | 0.37 ± 0.03 |
|  | BiKE + CBR (w/o GAT) | 18.15 ± 1.51 | 0.92 ± 0.03 | 37.10 ± 1.41 | 0.67 ± 0.03 | 46.70 ± 0.71 | 0.39 ± 0.03 |

**Table 11:** Test-set performance for *TWC in-distribution* (**IN**) and *out-of-distribution* (**OUT**) games using entities as context selectors.

## H  CASE-BASED REASONING AND OUT-OF-DISTRIBUTION GENERALIZATION

Out-of-distribution generalization has recently fueled significant research effort and several datasets and approaches have been proposed in the past few years (Bahdanau et al., 2019; Keysers et al., 2020; Atzeni et al., 2021). Our experiments on *TWC* allowed assessing the hypothesis that case-based reasoning can be used to tackle out-of-distribution (OOD) generalization in text-based games. Table 12 shows the absolute OOD generalization gap of the different agents evaluated in our experiments. Note that this table does not report any new result, but it simply provides the absolute difference between the values in Table 1 and 2. We observe that the agents relying on CBR achieve a considerably better generalization performance out of the training distribution, almost comparable to the results obtained on the same distribution as the training data. In some cases, the normalized score achieved by the CBR agents in the out-of-distribution games equals the score obtained in the in-distribution games. This happens because case-based reasoning forces the agent to map contexts including entities that were not seen at training time to the most similar contexts in the CBR memory. CBR allows the agent to solve completely new problems and generalize by effectively retrieving past cases and mapping the retrieved actions from the training distribution to the most similar options in the OOD setting. Note that the only good OOD generalization gap for the agents that are not relying on CBR (the **#Steps** of the **Text** agent on the **Hard** level) is an artifact of the experiment, as all agents were limited to a maximum of 50 steps.

Figure 6 shows a nice example of the capability of the agent to generalize OOD. In this case, entity embeddings were used as the context representations, and we observe that entities that are not included in the training distribution are correctly mapped to the right cluster. This shows that the CBR approach learns effective and generalizable context representations based on the objective of the games. These representations are then used by the agent to select relevant experiences and map the actions used at training time to the most viable alternative in the OOD test set.

|  | Easy | | Medium | | Hard | |
|---|---|---|---|---|---|---|
|  | #Steps | Norm. Score | #Steps | Norm. Score | #Steps | Norm. Score |
| **Text** | 6.07 | 0.10 | 1.82 | 0.05 | 0.16 | 0.10 |
| **TPC** | 7.15 | 0.11 | 2.28 | 0.04 | 1.55 | 0.13 |
| **KG-A2C** | 6.24 | 0.06 | 1.44 | 0.03 | 2.00 | 0.11 |
| **BiKE** | 7.32 | 0.11 | 1.67 | 0.03 | 2.81 | 0.11 |
| **CBR-only** | 1.30 | 0.00 | 0.27 | 0.01 | 0.59 | 0.02 |
| **Text + CBR** | 3.38 | 0.04 | 1.22 | 0.00 | 0.78 | 0.02 |
| **TPC + CBR** | 2.09 | 0.02 | 0.25 | 0.01 | 0.29 | 0.03 |
| **KG-A2C + CBR** | 2.30 | 0.05 | 0.89 | 0.02 | 0.99 | 0.02 |
| **BiKE + CBR** | 1.43 | 0.02 | 0.21 | 0.00 | 0.70 | 0.02 |

**Table 12:** Absolute out-of-distribution generalization gap in *TWC*

# I  ADDITIONAL BASELINES ON JERICHO

Several methods have been proposed recently for text-based games. In order to keep the results in the main paper more compact, we only included in Table 3 well-known and top-performing baselines that were evaluated on the full (or almost) set of *Jericho* games. For completeness, this section compares our best agent (**BiKE + CBR**) with the following additional methods:

- **Q*BERT** (Ammanabrolu et al., 2020) is a deep reinforcement learning agent that plays text games by building a knowledge graph of the world and answering questions about it;
- **Trans-v-DRRN** (Xu et al., 2020) relies on a lightweight transformer encoder to model the state of the game;
- **DBERT-DRRN** (Singh et al., 2021) makes use of DistilBERT (Sanh et al., 2019) fine-tuned on an independent set of human gameplay transcripts.

Table 13 shows the scores obtained by these baselines compared to our agent enhanced with CBR. Overall, we observe that the **BiKE + CBR** agent outperforms the baselines on the majority of the games, confirming the effectiveness of case-based reasoning as a viable approach to boost the performance of text-based RL agents.

| Game | Q*BERT | Trans-v-DRRN | DBERT-DRRN | BiKE + CBR |
|---|---|---|---|---|
| 905 | - | - | - | 0 |
| acorncourt | - | 10 | - | **12.2** |
| adventureland | - | 25.6 | - | **27.3** |
| afflicted | - | 2 | - | **3.2** |
| awaken | - | - | - | 0 |
| detective | 246.1 | 288.8 | - | **326.1** |
| dragon | - | - | - | 8.3 |
| inhumane | - | - | **32.8** | 24.2 |
| library | 10.0 | 17 | 17 | **22.3** |
| moonlit | - | - | - | 0 |
| omniquest | - | - | 4.9 | **17.2** |
| pentari | 48.2 | 34.5 | - | **52.1** |
| reverb | - | **10.7** | 6.1 | 6.5 |
| snacktime | - | - | 20 | **22.1** |
| temple | 7.9 | 7.9 | **8** | 7.8 |
| ztuu | 5 | 4.8 | - | . **87.2** |
| advent | - | - | - | 62.1 |
| balances | 9.8 | - | - | **11.9** |
| deephome | **1** | - | - | **1** |
| gold | - | - | - | 2.1 |
| jewel | - | - | **6.5** | 6.4 |
| karn | - | - | - | 0 |
| ludicorp | 17.6 | 16 | 12.5 | **23.8** |
| yomomma | - | - | 0.5 | **1** |
| zenon | - | - | - | 4.1 |
| zork1 | 33.6 | 36.4 | **44.7** | 44.3 |
| zork3 | - | 0.19 | 0.2 | **3.2** |
| anchor | - | - | - | 0 |
| enchanter | - | 20.0 | - | **36.3** |
| sorcerer | - | - | - | 24.5 |
| spellbrkr | - | 40 | 38.2 | **41.2** |
| spirit | - | - | 2.1 | **4.2** |
| tryst205 | - | 9.6 | 9.3 | **13.4** |

**Table 13:** Average raw score on the *Jericho* games. Results are taken from the original papers or "−" is used if a result was not reported.

# J  HYPERPARAMETERS AND REPRODUCIBILITY

All CBR agents are trained using the same hyperparameter settings and the same hardware/software configuration. As mentioned in Section 4, we use a pre-trained BERT model (Devlin et al., 2019) to represent initial node features in the state graph. BERT is only used to compute the initial representations of the entities and is not fine-tuned. We use the following hyperparameters for our experiments.

- We set the hidden dimensionality of the model to $d = 768$ and we use 12 attention heads for the graph attention network, each applied to 64-dimensional inputs.

- We use $n_l = 2$ seeded GAT layers for *TWC* and $n_l = 3$ for Jericho.

- On both datasets, we apply a dropout regularization on the seeded GAT with probability of $0.1$ at each layer.

- Similarly, for the experiments on *TWC*, we only sample the most recent context-action pair from $\mathcal{T}$, whereas we sample $k = 3$ pairs for Jericho. We used $k = 2$ for the scalability analysis depicted in Figure 5.

- The retriever threshold is kept constant to $\tau = 0.7$ across all experiments.

- On *TWC*, we train the agents for 100 episodes and a maximum of 50 steps for each episode. On Jericho, as mentioned, we follow previous work and we train for $100\,000$ valid steps, starting a new episode every 100 steps or whenever the games ends.

- We set the discount factor $\gamma$ to 0.9 on all experiments.

- For the ablation study on memory access, we set the output dimensionality of the RP and SRP methods to $p = 64$. For LSH, we set the number of hash tables to $l = 16$ and the length of the hash codes of $h = 8$ bits. We artificially limit the size of each bucket to the $4$ most recent entries.

Experiments were parallelized on a cluster where each node was dedicated to a separate run. The configuration of the execution nodes is as reported in Table 14.

| Resource | Setting |
|----------|---------|
| CPU | Intel(R) Xeon(R) CPU E5-2690 v4 @ 2.60GHz |
| Memory | 128GB |
| GPUs | 1 x NVIDIA Tesla k80 12 GB |
| Disk1 | 100GB |
| Disk2 | 600GB |
| OS | Ubuntu 18.04-64 Minimal for VSI |

**Table 14:** Hardware and software configuration used to train the agents

