# OpenReview forum: "Case-based reasoning for better generalization in textual reinforcement learning"
_ICLR.cc/2022/Conference — ICLR 2022 Poster_

### Official Review · Reviewer_HscQ · 2021-10-30

**Correctness:** 4
**Technical Novelty And Significance:** 2
**Empirical Novelty And Significance:** 2
**Recommendation:** 6
**Confidence:** 3

**Main Review:**

Strengths:
1. Authors show that combining CBR with existing RL methods improves agent performance significantly.
2. Authors conduct detailed experiments (existing RL models + CBR) which show the effectiveness of the proposed approach. The proposed model is more sample efficient and generalizes better.

Weaknesses:
1. The different components proposed in the paper, (e.g., CBR, KG) are not novel per se for the text-based RL agents. The paper has limited novelty.
2. There has been recent work that has used transformer-based LMs for text-based RL agents [e.g., 1,2,3]. Since Transformers are trained on very large corpora, they have been shown to encode world knowledge and hence work well in text-based world settings when combined with an RL agent. The authors do not compare to these approaches. It would be nice to have a comparison with these knowledge-based approaches as well.

[1] Deep reinforcement learning with transformers for text adventure games, Y. Xu, et al., 2020: https://ieeexplore.ieee.org/document/9231622
[2] Pre-trained Language Models as Prior Knowledge for
Playing Text-based Games, Singh, et al., 2020: https://arxiv.org/abs/2107.08408
[3] Stabilizing transformers for reinforcement learning. Parisotto, et al., 2020: https://arxiv.org/abs/1910.06764

Suggestion:

Authors say that they use A2C as the RL agent, however, they cite the paper that describes A3C, so it is not exactly clear if they are using a synchronous or asynchronous version of the Advantage Actor-Critic agent. More details on this would help.

**Summary Of The Paper:**

In this paper, the authors propose to improve the existing policy-based RL algorithms for the text-based world environments by incorporating knowledge via a case-based reasoning module. This improves out of distribution performance. The key idea in Case-Based Reasoning is to collect interactions that led to positive rewards in the past and try to map a novel situation to one of these past interactions to decide on the action in the current situation. In particular, authors represent the state of a text-based game by a knowledge graph and use message propagation (focussed on the sub-set of the nodes) to get the most similar representation of a new context.

**Summary Of The Review:**

The paper proposes the use of Case-Based Reasoning combined with Knowledge Graphs for improving RL agents for text-based environments. However, the approach has limited novelty as similar approaches have been proposed in the past. On the evaluation side, authors should also compare with other approaches (e.g., based on Transformers) that make use of external knowledge to generalize better.

---

> ### Author Response · Authors · 2021-11-22
> **Reply to Reviewer HscQ**
>
> We thank the reviewer for the feedback and for the time spent on our manuscript. We reply to the two main concerns in your review below.
>
> **Novelty.** To the best of our knowledge, we believe that case-based reasoning has not been explored to train RL agents in the way proposed in our  paper. This has also been mentioned by the other reviewers. _“I think this paper is exploring a novel direction, has good empirical results and would be a valuable addition to the literature”_ (**Reviewer Da5u**); _“As the related work shows, CBR to speed up RL has been tried before, however those methods are very different and not meant for the modern scale of deep RL [...] It would be beneficial to get RL practitioners to seriously consider adding CBR type approaches as a new type of strategy for boosting Deep RL”_ (**Reviewer aw9P**); _“The core idea is well defined and motivated: let’s get agents to use previous useful experiences to better improve the current policy. [...] The paper [...] presents a useful paradigm for thinking about text game agents and the results are interesting”_ (**Reviewer eX6x**). Therefore, we politely disagree on the lack of novelty of our approach.
>
>
> **Comparison to additional baselines.** Thanks for the references to additional baselines relying on transformer-based LMs for Jericho. We have included a comparison with the models that you suggested in **Appendix I** (**Table 13**). Note  that our approach outperforms Trans-v-DRRN and DBERT-DRRN in the vast majority of the games. Moreover, Trans-v-DRRN fails to perform better than the baselines in Table 3 on 12 out of  the 15 games it was evaluated on. Similarly, DBERT-DRRN only performs better than the baselines on 4 games out of the 14 games for which the paper reports results.
>
> Anyway, we agree that more analyses are beneficial to the reader to have a sense of how the model works and where the performance gains come from. Hence, we performed several additional experiments and ablation studies. As an example, **Appendix B** shows the results obtained by different baselines when enhanced with CBR on a subset of the Jericho games. **Appendix C, D, E, and F** further discuss several additional experiments.
>
> We hope that these additional analyses and our response will clarify some concerns and that you will reconsider your evaluation of our work.
>
> Minor comment: thanks for spotting the wrong reference to the A3C method. We fixed it!

---

### Official Review · Reviewer_eX6x · 2021-11-05

**Correctness:** 3
**Technical Novelty And Significance:** 3
**Empirical Novelty And Significance:** 3
**Recommendation:** 8
**Confidence:** 5

**Main Review:**

Strengths:

- The core idea is well defined and motivated. "Lets get agents to use previous useful experiences to better improve the current policy"

- The paper is well written overall I was mostly able to follow along, model design choices are mostly explained.

- The paper compares to multiple existing methods and the line up of related work baselines strengthens the work.

Clarifications/Concerns/Weaknesses:

- Section 3&4 in particular are a bit unclear and can be consolidated more. I think providing details of each portion under the relevant subsection of the 4 step CBR algorithm would make it a lot more clear.

- The retrieve portion in particular seems very similar to many other retrieval methods recently seen in text based game literature and the wider NLP community. It would be nice to see a comparison/ablation in *just the retrieve portion* to not use the knowledge graph and only text on *Jericho* (one obvious way would be to frame it as a Machine Reading Comprehension task to retrieve passages as in Guo et al. 2020 https://arxiv.org/abs/2010.02386 [which is cited and compared to for TWC]). Without such experiments it is relatively unclear whether the portions shown in Sec 4.1 are aiding, and if so how.

- The "Baseline agent + CBR" comparisons are good to have but would probably be more useful to have in Jericho than TWC. The main reason being that TWC is a singular "home" domain in which the commonsense knowledge (the text descriptions + genre knowledge) are more likely to be uniform throughout than with the varying genres like Jericho. This tests the limits of the CBR process and gives you the ability to analyze exactly in what portion of cases in certain types of knowledge transferable. In my opinion, such an analysis would likely prove to be the most valuable contribution of a work like this.

**Summary Of The Paper:**

The paper presents, at its core, a case-based reasoning (CBR) centered approach to solving text-based games. The CBR process shown is a 4 step process consisting of: retrieve (useful past experiences), reuse (the past experiences in a meaningful manner), revise (modify the current policy to account for prior experiences), and retain (decide which experiences to keep). This method is applied to an A2C agent that uses a knowledge graph based state representation, though it can be used along with other types of text game agents too.


**Summary Of The Review:**

The paper is well written and presents a useful paradigm for thinking about text game agents and the results are interesting. The main thing holding the paper back is the lack of analysis/ablations on certain portions of the proposed algorithm that make it unclear where the gain are coming from.


====Rebuttal Update====
I've read the rebuttal across the reviewers and am satisfied enough to increase my score.

---

> ### Author Response · Authors · 2021-11-22
> **Reply to Reviewer eX6x**
>
> Thanks for the positive comments and constructive suggestions. We really appreciate reviews that can result in an opportunity to improve the paper with additional experiments. We reply to your main points below.
>
> > The "Baseline agent + CBR" comparisons are good to have but would probably be more useful to have in Jericho than TWC. [...]
>
> Thanks for this comment. Following your suggestion, we included an experiment where we compare several baseline agents to their counterparts enhanced with CBR on a subset of the Jericho games. The results show that CBR consistently boosts the performance of the baseline agents on Jericho as well. More details are provided in **Appendix B** (**Table 5**)
>
> >  [...] It would be nice to see a comparison/ablation in just the retrieve portion to not use the knowledge graph and only text on Jericho [...]
>
> We liked the suggestion to include a study where we replace the graph-based retriever with a text-based approach similar to Guo et al. 2020 (note that we are already comparing to this approach for Jericho, not for TWC). The ablation study (described in **Appendix F**) shows that the retriever that does not use the knowledge graph (but only text) achieves slightly worse performance than our original implementation. The results of the experiment are reported in **Table 10**. Please see the updated version of the paper for more details.
>
> Finally, we would like to point out that, following the suggestion of the other reviewers, we performed several additional analyses and ablations. For instance, in **Appendix C** we investigate alternative techniques for accessing the memory and **Appendix D** discusses alternative methods to select which context-action pairs should be retained. We believe these analyses could be of interest to you as well and should make it clear where the performance gains in our approach are coming from.
>
> We appreciated the overall positive feedback on our work. Since you thought the main thing holding the paper back was the lack of analysis/ablations on portions of the proposed algorithm, we hope that our new analysis allays your concerns and you would reconsider your evaluation of our work.

---

> ### Author Response · Authors · 2021-11-26
> **Thanks again**
>
> Dear Reviewer,
>
> Thanks again for your time and for appreciating our paper. We hope you had a chance to look at our rebuttal. We would like to know whether our response and the new experimental results are sufficient to address your concerns. Are there any remaining points that you would like to discuss?
>
> Thanks again for your consideration,
>
> Authors of the submission

---

### Official Review · Reviewer_aw9P · 2021-11-06

**Correctness:** 3
**Technical Novelty And Significance:** 3
**Empirical Novelty And Significance:** 3
**Recommendation:** 8
**Confidence:** 4

**Main Review:**

Using the Neurips rubric:

Originality: As the related work shows,  CBR to speed up RL has been tried before, however those methods are very different and not meant for the modern scale of deep RL and contextualized representations+retrieval methods. The actual details of the CBR model used in this paper, seeded graph attention and vector quantization to aid retrieval seem relevant at least in their application to doing CBR for RL.
The basic experiments are straightforward, applying combinations of CBR and sota RL methods on the 2 TAG domains. However, there are a couple of novelties in the ablations and a very good qualitative analysis to show where the gains from CBR are coming from (fig 3).

Quality: I thought the model described in sec 4 was interesting and mostly well-motivated. Some minor questions:
  a. comparing the 2 equations in seeded graph attention, it looks like the h's dependency on $\alpha$ is quadratic, which seems unusual for attention?
b. why sum for the final representation and not average? it could lead to wierd biases when the number of entities are different between states.
c. The VQ approach is interesting, but I am a bit skeptical that it is necesary. Very efficient methods exist to do retrieval in large dim spaces, with some effort. So for practical applications, it might just be easier to do that. Can you share some quantitative measurements to show that this code splitting approach is necessary?
d.  The claim that CBR lets you generalize OOD seems not supported and my intuition is almost the opposite. Can you elaborate?
e. The CBR and neueral agent seem to be separately trained. Can they be jointly optimized or at least can the neural agent be trained with  some knowledge of when the CBR will over-ride it so that it's policy can adapt to that?
f. Fig 4: it would be more interesting if this could show the "counterfactual" probabilties. since the neural agent always loses to the CBR in action selection, what if the neural agent had been allowed to execute, would it still have been successful?
g. Is locality sensitive hashing another way to do efficient retrieval that can generalize?

Clarity: the paper is well written and all major parts are well-explained. A few details could use some clarity:
a.  when you say "the final set of all retrieved actions and the corresponding relevance", can you be more precise and define the tuple or whatever exactly the data structure is.
b. sec 4.3: "applied to the entities": how does it choose which entities to use?
c. Please elaborate on the Adolphs and Hofmann method, even though it's not an original contribution it is a core part of your system.
d. The jump from 24% to 73% in win rate on Jericho seems really big ! You should highlight it more.

**Summary Of The Paper:**

This paper describes how to apply  a combination of case-based reasoning and RL methods to improve performance of agents on text-adventure game type tasks. It introduces a GNN representation of state and a vector-quantized encoding scheme so that contextual information about successful actions from the past can be retrieved and re-used. Experiments show a significant increase in performance, along with ablations showing value of the CBR  especially on OOD environments.Finally there is qualitative insights  of the memory representation showing how the CBR helps.

**Summary Of The Review:**

Fairly well-thought out scalable approach to adding CBR to a tough RL setting which has got some attention recently. Strong results and good empirical analysis. It would be beneficial to get RL practitioners to seriously consider adding CBR type approaches as a new type of strategy for boosting Deep RL.

---

> ### Author Response · Authors · 2021-11-22
> **Reply to Reviewer aw9P**
>
> Thanks for the overall positive feedback and the valuable comments. We appreciate your suggestions and the opportunity to further improve the paper.  We reply to your main questions below.
>
> > **a.** Comparing the 2 equations in seeded graph attention, it looks like the h's dependency on $\alpha$ is quadratic [...]
>
> The $\alpha$ in the first equation is only meant to do a weighted average over the $\beta$ coefficients of the neighbors. This is inspired by the similar architecture of Sun et al. [1], who used a directed message propagation for open-domain question answering and employed the attention coefficients in the same way.
>
> > **b.** Why sum for the final representation and not average? it could lead to wierd biases when the number of entities are different between states.
>
> We only sum on the representations of the seed entities. These entities are taken from each admissible action separately, to produce different action-specific representations. Hence, in practice, the number of entities is only 2 or 1, and we pass the result to a FFN.
>
> > **c.** The VQ approach is interesting, but I am a bit skeptical that it is necessary. [...] Can you share some quantitative measurements to show that this code splitting approach is necessary?
>
> Thanks for the valuable suggestion! We performed an ablation study where we compare the vector quantization with other techniques, including random projection, sign random projection and locality sensitive hashing. The results of the experiment are reported in **Appendix C** (**Table 6**) and show that our VQ-based implementation achieves better results compared to the other approaches.
>
> > **d.** The claim that CBR lets you generalize OOD seems not supported and my intuition is almost the opposite. Can you elaborate?
>
> Thanks for this comment. We have made this point more clear by adding an Appendix that explains why case-based reasoning obtains good OOD generalization. In short, the reason is that it forces the agent to map contexts from the OOD test set to the most similar contexts in the training set and effectively reuses the retrieved experiences to detect the most viable valid action in the OOD setting. More details are provided in **Appendix H**.
>
> > **e.** The CBR and neueral agent seem to be separately trained. Can they be jointly optimized [...]?
>
> Yes, we had already tried training the CBR retriever and the neural agent jointly. However, this does not yield any performance gain and requires changing the architecture of the agent to make it aware of the action candidates produced by the CBR, so that this information can be included in the action scoring mechanism. For these reasons, we switched to the implementation described in the main paper. We have included in **Appendix E** the results obtained by optimizing the retriever and the neural agent jointly, both on TWC (**Table 8**) and Jericho (**Table 9**)
>
> > **f.** Fig 4: it would be more interesting if this could show the "counterfactual" probabilties. [...] what if the neural agent had been allowed to execute, would it still have been successful?
>
> Thanks for this suggestion. We have incorporated the analysis you suggested in Figure 4.
>
> > **g.** Is locality sensitive hashing another way to do efficient retrieval that can generalize?
>
> Yes, see the reply to point c and Appendix C for more details.
>
> **Clarity.** Thanks for the comments!
>
> > **a.** when you say "the final set of all retrieved actions and the corresponding relevance", can you be more precise [...]?
>
> Thanks for the suggestion, we described more formally the set of retrieved actions in Section 3 (although the precise data structure is already shown in Alg. 1)
>
> > **b.** sec 4.3: "applied to the entities": how does it choose which entities to use?
>
> In Sec 4.3, the reused action is applied to the same entities mentioned in the action used to build the context selector ($\mathcal{V}_{a_t}$).
>
> > **c.** Please elaborate on the Adolphs and Hofmann method, even though it's not an original contribution it is a core part of your system.
>
> The method used to train our model (from Adolphs & Hofmann) is explained in **Appendix A**.  We included a reference to Appendix A in Section 5.
>
> > **d.** The jump from 24% to 73% in win rate on Jericho seems really big ! You should highlight it more
>
> Thanks, we have highlighted this in Section 6.3!
>
> Thanks for appreciating our paper and our empirical results. Hope our response further convinces you about our work.
>
>
>
> [1] Haitian  Sun,  Bhuwan  Dhingra,  Manzil  Zaheer,  Kathryn  Mazaitis,  Ruslan  Salakhutdinov,  and William Cohen. Open domain question answering using early fusion of knowledge bases and text. EMNLP 2018.

---

> > ### Comment · Reviewer_aw9P · 2021-12-01
> > **Response**
> >
> >
> > I think HscQ's questions were good ones, and the authors seem to have addressed them by comparison with other transformer-based methods. It seems that their particular architecture in fig 1 (KG, seeded attention etc) is required to get the results they did.
> >
> > One remaining confusion I have is philosophical rather than technical. Revisiting the idea of CBR again as described in sec 2, how is that different from the generic notion of a policy? Is the author's method better seen as simply a new approach to representing policies in structured enviroments that take max advantage of textual representations in order to do better IND and OOD generalization?
> >
> > The authors have done a pretty exhaustive job addressing other concerns, I am raising my score by 1.

---

> ### Author Response · Authors · 2021-11-26
> **Thanks again**
>
> Dear Reviewer,
>
> Thanks again for your time and for appreciating our paper. We hope you had a chance to look at our rebuttal. We would like to know whether our response and the new experimental results are sufficient to address your concerns. Are there any remaining points that you would like to discuss?
>
> Thanks again for your consideration,
>
> Authors of the submission

---

> ### Author Response · Authors · 2021-11-29
> **Kind reminder**
>
> Thanks again for your valuable feedback and your positive review. As today is the end of the final stage of the discussion, we would be glad if you could acknowledge that you have read our rebuttal. We believe that we have comprehensively addressed all your comments, but we remain available in case you have any further inquiry. Thanks!

---

### Official Review · Reviewer_Da5u · 2021-11-06

**Correctness:** 4
**Technical Novelty And Significance:** 3
**Empirical Novelty And Significance:** 3
**Recommendation:** 8
**Confidence:** 4

**Main Review:**

Strengths:
- Solid approach combining CBR and RL
- Experiments are thorough and convincing.
- Writing is quite clear

Weaknesses:
- Some design choices may need more justification/analysis (see below, mainly points 3 and 4)

Comments:
1. In the retriever module, how is the threshold $\tau$ chosen?
2. Algorithm 1 describes performing retrieval using multiple context selectors $\mathcal{C}_t$. How are these specified and how are they different from each other? Do they store different types of contexts?
3. The ‘Retain’ module seemed a bit ad-hoc to me since the module stores actions with positive rewards only. This goes against the key premise of performing good credit assignments in RL since the crucial action that led to this positive reward may occur 5 or 10 or even 20 steps before the transition with the positive reward. I suspect this is why storing previous actions helps empirically. So, to me, this design choice seems arbitrary and not well motivated. In fact, why not store negative actions as well or even use a prioritization scheme based on TD error similar to that in https://arxiv.org/abs/1511.05952 ? An ablation on this (even on a small subset of the games) may be helpful.
4. The context discretization scheme in 4.2 reminds me of this paper (https://arxiv.org/abs/2103.13552) that used hash functions to represent the state. Would just hashing the state (e.g. with locality sensitive hashing (LSH) on fixed pre-trained BERT representations) with a random function work as well as the learned discretization scheme? Again, an ablation on just a subset of the games might be useful here.
5. On the same point as above, the paper mentions that the discretization scheme helps tackle the issue of changing representations over the course of training. However, since the discretization is also a learned function, wouldn’t that also potentially change over time? It wasn’t clear how this solves the issue of the same context mapping to two different entries in the memory.
6. In table 3, the term “win rate/win count” for the last row was confusing. I took it to be the win rate on the games themselves, but it seems to be a comparison across the different models. I suggest changing it to something clearer.
7. It is also unclear how exactly the model chooses between two actions which have the same template, since they have the same relevance $\delta$ value (which depends only on the template?). For example, in figure 3, I’m not sure how the agent picks `Put emerald in case` over `Put jewels in case`.

**Summary Of The Paper:**

This paper proposes an approach to combine case-based reasoning with reinforcement learning for text-based games. The method works by keeping track of states that received positive rewards in the game, and then having a retrieval mechanism to retrieve similar contexts at a new state in the game. The authors use a quantization technique for efficient storage and retrieval, and a fallback neural policy (trained with RL) in cases where none of the retrieved action templates result in an admissible action in the current state. The writing is quite clear and the experiments are mostly convincing, barring a few questions I have.

**Summary Of The Review:**

I think this paper is exploring a novel direction, has good empirical results and would be a valuable addition to the literature. Addressing the points mentioned above would further strengthen the paper and I’m happy to raise my score if the authors can provide convincing responses.

---

> ### Author Response · Authors · 2021-11-22
> **Reply to Reviewer Da5u**
>
> We thank the reviewer for the thorough review and the valuable suggestions. We reply to each of your questions below. Also, please note that, besides the ablation studies you suggested, we performed several additional experiments that may still be of interest to you. Please see the updated version of the paper for more details.
>
> > **1.** In the retriever module, how is the threshold $\tau$ chosen?
>
> The main purpose of the threshold $\tau$ is to avoid retrieving irrelevant context-action pairs from the memory (e.g., in the early training stages). In order to avoid this issue, we set $\tau$ to a value that implies a large similarity between the query context and the retrieved ones. We opted to set $\tau = 0.7$ based on preliminary experiments on a subset of the TWC games.
>
> > **2.** Algorithm 1 describes performing retrieval using multiple context selectors. How are these specified and how are they different from each other? [...]
>
> Context selectors need to be action-specific, as they are meant to represent the portion of the state that is relevant to the execution of a given action. Hence, we compute a different context representation for each valid action (we assume that the set of valid actions is given at each time step). In our implementation, this is achieved using a graph attention seeded at the entities that the action is applied to.
>
> > **3.** The ‘Retain’ module seemed a bit ad-hoc to me since the module stores actions with positive rewards only. [...] Why not [...] use a prioritization scheme based on TD error [...]?
>
> Thanks for this very valuable suggestion. Indeed, our implementation is carefully designed for TBGs and works well for our use case, showing that in this context the main challenge lies in grounded language understanding and in the accurate modeling of the state of the game. However, we agree with you that other implementations of the retain module are possible. While the architecture described in Section 4 is specific to our use case, the algorithm described in Section 3 aims to be more general. Therefore, following your suggestion, we slightly changed the retain step to use a $\textit{retain}$ function that selects which of the previous actions should be retained. Both our original implementation (storing the last $k$ actions) and your suggestion fall into this general formulation. Then, we performed an ablation study, where we compared different versions of the retain module. One of these variants samples the context-action pairs to be retained according to the TD error as you suggested. Overall, we found that our original implementation works best in practice, but the other variants also obtain competitive results. The results of the ablation study are reported in **Appendix D** (**Table 7**).
>
> > **4.** [...] Would just hashing the state (e.g. with LSH [...] work as well as the learned discretization scheme? [...]
>
> Thanks again for another valuable comment! We performed an ablation study where we replaced the vector quantization with other techniques, like LSH, random projection and sign random projection. The results are reported in **Appendix C** (**Table 6**). Overall, we found that the vector quantization achieves better performance, but LSH also obtains good results. However, note that LSH requires storing the full continuous context representations as well, in order to detect false positives falling in the same bucket as the query context, but having small cosine similarity with it. This makes vector quantization a more efficient alternative.
>
> > **5.** [...] The paper mentions that the discretization scheme helps tackle the issue of changing representations over the course of training. However, [...] wouldn’t that also potentially change over time? [...]
>
> The discretization scheme helps to tackle the issue of representations changing over time, but the other reason why this works in practice is that the retriever is pretrained, as mentioned in Section 5. This minimizes large fluctuations in the continuous context representations and the discretization function further cancels small variations.
>
> > **6.** In table 3, the term “win rate/win count” for the last row was confusing [...]
>
> The terminology is taken from Guo et al. (2020). We agree that it can be confusing and we changed it in the updated version of the paper.
>
> > **7.** It is also unclear how exactly the model chooses between two actions which have the same template, since they have the same relevance [...]
>
> The relevance $\delta$ does not depend on the template. It depends on the similarity between the context representation stored in the memory and the current context. In the example in Figure 3, the agent picks the action “Put emerald in case” because the context in which that action was executed is more similar to the current context, where the action that will give positive reward is “Put diamond in case”.
>
> Thanks for appreciating our paper. Hope our response further convinces you about our work.

---

> > ### Comment · Reviewer_Da5u · 2021-11-22
> > **Thanks for the clarifications**
> >
> > Thank you for your responses, which help answer my questions, and I encourage you to add in these additional clarifications to the paper. I also appreciate the additional experiments in the Appendix, which further strengthens the paper's findings and provides better perspective to evaluate the design choices made in the paper. Overall, I'm happy to raise my score!
> > One last suggestion I have with respect to the 'Best Agent' row in Table 3 is to discount games where none of the agents get off the ground and get 0 scores or all of them basically get the same score (e.g. 1). This might be a more accurate way to convey the performance comparison between the different agents.

---

> > > ### Author Response · Authors · 2021-11-26
> > > **Thanks for your suggestions and review**
> > >
> > > Thanks for appreciating our work. We will make the changes you suggested in the next version.

---

### Author Response · Authors · 2021-11-22
**General comments**

We thank all the reviewers for the overall positive feedback, the valuable suggestions, and the time spent on our manuscript. Following your comments, we believe we have significantly improved the paper with additional ablation studies and experiments. The main improvements we made to the paper are the following.

* We investigated the performance gains obtained by different baseline agents when enhanced with case-based reasoning on the Jericho environment (**Reviewer eX6x**). The results (**Appendix B**, **Table 5**) show that CBR consistently boosts the performance of the neural agents, similarly to what we observed in TWC.
* We evaluated different techniques to efficiently retrieve previous experiences from the case memory, including locality sensitive hashing, random projection, and sign random projection (**Reviewers Da5u and aw9P**). We found that the original VQ-based approach performs best and is more efficient (see **Appendix C**, **Table 6**).
* We assessed alternative options for selecting which actions should be retained in the memory, such as sampling actions based on the TD error (**Reviewer Da5u**). We observed that the method based on the TD error could be a viable alternative, but our original implementation achieves the best results (see **Appendix D**, **Table 7**).
* We reported the results obtained by a version of the CBR method that trains the agent and the retriever jointly (**Reviewer aw9P**). This requires changing the architecture of the neural agent and does not yield any noticeable improvement (see **Appendix E**, **Tables 8 and 9**)
* We replaced our graph-based retriever with a multi-paragraph text-based implementation (as in Guo et al. 2020) to assess the effectiveness of the knowledge graph and the seeded graph attention (**Reviewer eX6x**). The results (**Appendix F**, **Table 10**) show that modeling the state as a knowledge graph allows better capturing contextual information compared to the text-based retriever.
* We compared to additional baselines for Jericho (**Reviewer HscQ**), confirming that our approach outperforms the references given by the reviewer (see **Appendix I**, **Table 13**).
* We clarified some points, added a discussion on out-of-distribution generalization (**Appendix H**, **Reviewer aw9P**), and made additional minor improvements.

Overall, **all the additional experiments and analyses we included in the paper confirmed the effectiveness of our implementation**. We highly appreciated the opportunity to improve our work and we would like to request the reviewers to have a look at the updated version of our paper. We are looking forward to your feedback on our additional experiments and we hope that our new analyses will further convince you about our work.

---

### Decision · Program_Chairs · 2022-01-20

**Decision:**

Accept (Poster)

**Comment:**

This paper describes how to apply a combination of case-based reasoning and RL methods to improve the performance of agents in text-adventure games.  The reviewers unanimously recommend acceptance.  This work is both insightful and practical.  This is a valuable contribution.  Well done!